# THE PROBE PARADIGM: A THEORETICAL FOUNDATION FOR EXPLAINING GENERATIVE MODELS

**Amit Kiran Rege**
Department of Computer Science
University of Colorado Boulder
Boulder, CO, USA
`amit.rege@colorado.edu`

## ABSTRACT

To understand internal representations in generative models, there has been a long line of research of using *probes* i.e. shallow binary classifiers trained on the model's representations to indicate the presence/absence of human-interpretable *concepts*. While the focus of much of this work has been empirical, it is important to establish rigorous guarantees on the use of such methods to understand its limitations. To this end, we introduce a formal framework to theoretically study explainability in generative models using probes. We discuss the applicability of our framework to number of practical models and then, using our framework, we are able to establish theoretical results on sample complexity and the limitations of probing in high-dimensional spaces. Then, we prove results highlighting significant limitations in probing strategies in the worst case. Our findings underscore the importance of cautious interpretation of probing results and imply that comprehensive auditing of complex generative models might be hard even with white box access to internal representations.

## 1 INTRODUCTION

Generative models have achieved remarkable success in generating high-quality data across various domains, such as text, images, and audio. Models like large language models (LLMs) (Radford et al., 2019; Brown et al., 2020), diffusion models (Ho et al., 2020), and generative adversarial networks (GANs) (Goodfellow et al., 2014) have demonstrated impressive capabilities in tasks ranging from text completion to realistic image synthesis. Despite these advancements, understanding the internal mechanisms and representations learned by these models remains a significant challenge.

Probing methods have emerged as a valuable approach to interpret the hidden representations of neural networks (Alain & Bengio, 2018; Belinkov, 2022). By training simple classifiers, known as *probes*, researchers can detect the presence of human-understandable concepts within the hidden layers of a model. These concepts can range from syntactic structures in language models to visual features in image generation models.

However, probing high-dimensional representations introduces challenges related to overfitting and the expressiveness of the probes. Expressive probes, such as multi-layer perceptrons (MLPs), can perfectly fit training data with random labels, leading to misleading conclusions about the model's internal representations (Zhang et al., 2017b). Conversely, simpler probes may fail to capture complex, non-linear relationships present in the data (internal representations in our case). Thus, it is necessary to strike a balance in order to use probes to understand a model's internal reasoning process.

On the other hand, there has been a flurry of work on explainability methods in the supervised, classification setting (Ribeiro et al., 2016; Lundberg & Lee, 2017). These methods, however, have been shown to be brittle (Alvarez-Melis & Jaakkola, 2018) so there has been some recent work on trying to provide *formal* guarantees for explanations (Yadav et al., 2023; Dasgupta et al., 2022; Bhattacharjee & von Luxburg, 2024). While probing techniques have been used in a wide variety of generative models empirically (Belinkov, 2022; Hewitt & Liang, 2019), a formal framework for their analysis is lacking.

In this paper, we initiate the study of theoretical properties of explainability in generative models using a general, model-agnostic formal framework using probes. Our framework formalizes the use of probes — auxiliary models trained to detect specific human-understandable concepts detected within the model's hidden representations. By maintaining generality, we aim to encompass a wide range of real-world probing-based explainability methods while providing a solid foundation for theoretical analysis.

To this end, we introduce the first formal framework utilizing probes in generative models, providing a rigorous foundation for explanations. Our key contributions are:

- A first formal framework to theoretically study the explainability of generative models through the lens of probing.
- Demonstration of our framework's versatility through application to diverse generative models, including language models, diffusion models, and GANs.
- Theoretical results on sample complexity for concept detection with linear probes and fundamental limitations in high-dimensional spaces.
- Theoretical validation of empirical findings from prior work about the overfitting risks of expressive probes in high-dimensional settings.
- Novel results revealing inherent limitations of probing internal representations, including challenges with fine-grained concepts and nonlinear concept detection.

Through this work, we aim to advance the theoretical understanding of explainability in generative models and offer a foundation for further research into offering rigorous guarantees for auditing and interpretability in such systems.

## 2 FORMAL FRAMEWORK

In this section, we present our formal framework to understand concepts learned in generative models using probes.

### 2.1 DEFINITIONS AND NOTATION

We assume generative models have a layered architecture where internal representations can be extracted at each layer. When we say "internal representations" in this paper, we take these to mean dense layer outputs, attention head outputs, context embeddings, or any other intermediate computation. This allows our framework to be extremely general.

**Definition 1** (Generative Model). *Let $M$ be a generative model with $L$ layers, denoted as $\{l_1, l_2, \ldots, l_L\}$. The model maps inputs $x \in \mathcal{X}$ to outputs $y \in \mathcal{Y}$, such that $y = M(x)$.*

**Definition 2** (Hidden Representations). *For an input $x \in \mathcal{X}$, the hidden representation at layer $l$ is denoted by $h_l(x) \in \mathcal{H}_l$, where $\mathcal{H}_l \subseteq \mathbb{R}^{d_l}$ is the hidden space associated with layer $l$ of dimensionality $d_l$.*

A common strategy to understand model internals is to train simple, shallow classifiers (probes) on these hidden representations to detect if a particular concept, such as "face" in visual generative models or "truthfulness" in language models, is present in the model's internal activations. Specifically, we define each concept at a layer $l$ of the model and train a binary classifier $p_{c,l}$ to tell us if the concept $c$ is present in the representation at that layer. Note that the choice of concept to test is inherently subjective and not amenable to formalization.

**Definition 3** (Concepts in Representation Space). *A concept $c \in \mathcal{C}$ is a human-understandable property we wish to detect in the model's hidden representations. For each layer $l \in \{1, 2, \ldots, L\}$, we define a probe $p_{c,l}$ that outputs the probability that concept $c$ is present in the hidden representation $h_l(x)$.*

Thus, the concept $c$ is detected at layer $l$ based on the probe $p_{c,l}(h_l(x))$, which returns a probability value in the range $[0, 1]$ indicating the confidence that concept $c$ is present in the hidden representation $h_l(x)$.

**Definition 4** (Probes). *A probe $p_{c,l}$ is a probabilistic binary classifier associated with concept $c \in C$ and layer $l \in \{1, 2, \ldots, L\}$. It outputs a probability score reflecting the likelihood that concept $c$ is present in the hidden representation $h_l(x)$.*

*The probe is a function $p_{c,l} : \mathcal{H}_l \to [0, 1]$, where:*

$$p_{c,l}(h_l(x)) = \mathbb{P}(\text{concept } c \text{ is present} \mid h_l(x)).$$

*We assume probes belong to some pre-specified hypothesis class $\mathcal{P}$.*

The probe is typically a linear classifier, such as a logistic regression, which outputs the probability of concept detection. The decision boundary for the concept $c$ is defined by a linear hyperplane in the representation space $\mathcal{H}_l$.

Typically, probes are trained by creating a dataset of $\{h_l(x_i), y_i\}$ where $y_i$ indicates the presence/absence of the concept being tested using a gold-standard annotated dataset. The exact procedure to do this depends on the modality of the data and architecture of the model (among other variables). In this paper, to remain model-agnostic, we will not concern ourselves with details involved in training such probes - we assume that such a dataset can be created and a probe can be trained.

**Definition 5** (Thresholded Concept Detection). *To convert the probabilistic output of the probe into a binary decision, we introduce a threshold $\tau \in [0, 1]$. The concept $c$ is detected at layer $l$ if:*

$$p_{c,l}(h_l(x)) > \tau.$$

If no threshold is explicitly chosen, a default value of $\tau = 0.5$ may be used to convert the probability into a binary decision.

## 2.2 EXAMPLES OF GENERATIVE MODELS WITHIN OUR FRAMEWORK

To illustrate the applicability of our framework, we provide examples of how different generative models fit within our formalization, mapping each component accordingly. These are meant to be fairly general to show the large scope of our framework but by no means are they exhaustive. We provide additional examples in Appendix B.

### 2.2.1 LARGE LANGUAGE MODELS (LLMs)

- **Generative Model:** The generative model $M$ is a large language model, such as GPT-3 (Brown et al., 2020) or BERT (Devlin et al., 2018), which generates text sequences given an input prompt.

- **Hidden Representations:** For an input sequence $x \in \mathcal{X}$, the hidden representation at layer $l$ is denoted by $h_l(x) \in \mathcal{H}_l \subseteq \mathbb{R}^{d_l}$, where $d_l$ is the dimensionality of the hidden states at that layer. These representations could be the activations after self-attention and feed-forward layers.

- **Concepts:** Concepts $c \in \mathcal{C}$ in the context of language models could include syntactic properties (e.g., part-of-speech tags), semantic roles (e.g., agent, patient), or stylistic attributes (e.g., formality level).

- **Probes:** Probes $p_{c,l}$ are classifiers trained to detect the presence of concept $c$ in the hidden representation $h_l(x)$. For example, a probe could be trained to predict the part-of-speech tag of a word based on its hidden representation at layer $l$.

### 2.2.2 DIFFUSION MODELS

- **Generative Model** The generative model $M$ is a diffusion model used for image generation, such as Denoising Diffusion Probabilistic Models (DDPM) (Ho et al., 2020), which generate images through an iterative denoising process.

- **Hidden Representations:** For an initial noise vector $x \in \mathcal{X}$, the hidden representation at diffusion step $l$ is $h_l(x) \in \mathcal{H}_l \subseteq \mathbb{R}^{d_l}$. These representations correspond to the intermediate noisy images or latent variables at each step of the diffusion process.

- **Concepts:** Concepts $c \in \mathcal{C}$ could include the presence of specific objects (e.g., "cat", "car"), styles (e.g., "impressionist", "photorealistic"), or attributes (e.g., color schemes, textures) in the images being generated.
- **Probes:** Probes $p_{c,l}$ are trained to detect whether a concept $c$ is present in the hidden representation $h_l(x)$ at diffusion step $l$. For example, a probe could predict whether the image at a certain diffusion step contains a particular object or style.

## 2.3 MAPPING COMPONENTS TO OUR FRAMEWORK

In all these examples, we can map the components of our framework as follows:

- **Generative Model** ($M$): The model responsible for generating data, such as text or images.
- **Hidden Representations** ($h_l(x)$): Intermediate activations or latent variables at different layers or steps within the model.
- **Concepts** ($c$): Human-understandable properties or attributes that we aim to detect within the model's representations.
- **Probes** ($p_{c,l}$): Classifiers trained to identify the presence of concepts within the hidden representations at specific layers or steps.

Our framework provides a unified approach to analyze and interpret various generative models by examining how concepts are represented internally at different levels of abstraction.

## 2.4 BASIC RESULTS

Typically, a practitioner will decide the concept they want to test for and then provide the model with a ground truth dataset to produce "good" internal representations. These are then recorded and used to train linear probes across all the layers in the model. If a concept cannot be detected at a particular layer, the probe will be as good as random guessing. In practice, it is not known which layer a particular concept exists at apriori, thus, probes are trained to detect a concept on all layers and the probe with the highest accuracy is chosen to study the concept of interest.

We begin our theoretical anlysis by first studying the number of samples required to learn a concept with high accuracy using this strategy (across all layers). All proofs (unless specified) are in Appendix A.

**Proposition 1** (Sample Complexity for Multi-Layer Concept Detection). *Let $M$ be a generative model with $L$ layers, $c$ a concept to detect, and $\mathcal{H}_l \subseteq \mathbb{R}^{d_l}$ the hidden space for layer $l$. Using linear probes $p_{c,l}$ to detect concept $c$ at each layer independently, and selecting the best performing probe based on a validation set, the total sample complexity to achieve an error of at most $\epsilon$ with probability at least $1 - \delta$ is:*

$$n = O\left(\frac{1}{\epsilon^2}\left(d_{\max} + \log\frac{L}{\delta}\right)\right),$$

*where $d_{\max} = \max_{l \in \{1,2,...,L\}} d_l$ is the maximum dimensionality across all layers.*

The theorem is a straightforward application of standard PAC learning results combined over all layers in the model.

Next, we attempt to understand the limitations of using linear probes in our framework. To do this, we examine the capacity of the representation space to encode independent concepts.

**Definition 6** (Independent Concepts). *Two concepts $c_1$ and $c_2$ are considered independent in a representation space $\mathcal{H} \subseteq \mathbb{R}^d$ if their corresponding linear probes $p_1$ and $p_2$ have weight vectors $w_1$ and $w_2$ that are linearly independent.*

This definition captures the idea that independent concepts should be detectable by probes that focus on different aspects of the representation space.

**Proposition 2** (Maximum Independent Concepts for Linear Probes). *Let $\mathcal{H} \subseteq \mathbb{R}^d$ be the hidden space for a layer in a generative model. The maximum number of mutually independent concepts that can be represented by linear probes is $d$.*

*Proof.* A linear probe $p$ for a concept $c$ in the hidden space $\mathcal{H}$ is represented as $p(h) = \sigma(w^T h + b)$, where $w \in \mathbb{R}^d$ is the weight vector and $b$ is a scalar bias. The decision boundary of this probe is the hyperplane $w^T h + b = 0$.

The maximum number of linearly independent weight vectors $w$ in $\mathbb{R}^d$ is $d$. This directly corresponds to the maximum number of linearly independent hyperplanes that can be defined in $d$-dimensional space, which in turn represents the maximum number of independent concepts detectable by linear probes. $\square$

This theorem has several implications for our framework. It establishes an upper bound on the number of independent concepts detectable using linear probes in a $d$-dimensional hidden space, informing the expressiveness of our representation space and the limitations of our probing approach. When trying to look for multiple concepts within a given representation space, we must consider that beyond $d$ concepts, some level of dependence or overlap is guaranteed. This result also guides the required dimensionality of the hidden space for representing a given number of independent concepts i.e. a larger representation space potentially allows for more non-overlapping concepts to be present.

Such limitations in expressivity have led to some prior work considering using more expressive probes such as MLPs (Hewitt & Liang, 2019; Belinkov, 2022). The idea is that certain complex concepts will only show up non-linearly in the representation space. However, as we prove in the next section, this can cause the issue of now having to decipher whether the probe is actually showing us something meaningful about the representation space or just very good at the getting a high accuracy. Besides, the entire point of using probes was to have explainability - if we use MLPs, one has to now presumably explain the MLPs themselves as well.

## 2.5 LIMITATIONS OF PROBING IN HIGH DIMENSIONS

We now present a result that formalizes the phenomenon of probes fitting random labels in high-dimensional spaces. We show that given sufficiently expressive probes (such as MLPs), a practitioner examining a model can be mislead into thinking that the model encodes a concept when it does not. The proof goes through via standard concentration of measure arguments.

**Theorem 1** (Limitations of Probing in High Dimensions). *Let $M$ be a generative model with $L$ layers, and let $h_l(x) \in \mathbb{R}^d$ be the hidden representation at layer $l$. Let $\mathcal{C}$ be a set of concepts, and $\mathcal{P}$ be a set of probes including sufficiently expressive neural networks. Assume that $h_l(x)$ is drawn from a standard normal distribution $\mathcal{N}(0, I_d)$.*

*For any $\epsilon > 0$ and $\delta > 0$, there exists a dimension $D = O(\frac{1}{\epsilon^2} \log \frac{1}{\delta})$ such that for all $d > D$, the following holds with probability at least $1 - \delta$ over the choice of a random concept $c$ (i.e., a random labeling):*

1. *There exists a probe $p \in \mathcal{P}$ that achieves perfect training accuracy on $n = \Omega(d)$ samples: $PERF_{train}(p, h_l, c) = 1$.*

2. *The expected test accuracy of $p$ is $PERF_{test}(p, h_l, c) \leq 0.5 + \epsilon$.*

3. *The mutual information $I(c; h_l) \leq \epsilon$.*

This result demonstrates that in high-dimensional spaces, expressive probes can perfectly fit random labels on training data, yet fail to generalize. The representations do not meaningfully encode the concept, as evidenced by the negligible mutual information. Thus, it underscores the importance of cautiously interpreting high probe accuracies as it could be a result of overfitting in a high-dimensional space.

Indeed, several works (Hewitt & Liang, 2019) have in the past demonstrated spurious correlations found using MLP probes and have suggested the use of regularization or training "control" probes acting as baselines trained on random labels. While these are valuable, Theorem 1 shows they might not be enough, as both real concepts and random controls can be "detected" in high-dimensional spaces. This finding has been validated in prior work where the difference in accuracy of probes trained on actual representations and random ones is small for larger, more expressive probes (Hewitt & Liang, 2019).

Thus, we are faced with a trade-off - Linear probes have lower risk of overfitting but may miss complex concept encodings while MLP probes can detect more complex relationships but are at higher risk of fitting noise.

## 3 LIMITATIONS OF PROBING INTERNAL REPRESENTATIONS

Building on our framework and initial results, we now turn to a deeper investigation of the fundamental limitations of probing techniques. While the previous section established the basic machinery for understanding probes and demonstrated some practical challenges in high-dimensional spaces, this section reveals additional, concrete theoretical barriers to concept verification in generative models. We present a series of impossibility results that highlight inherent limitations in our ability to verify certain types of concepts, regardless of the probing strategy employed.

Our analysis focuses on three key aspects: the verification of fine-grained concepts, the limitations of linear probes for nonlinear concepts, and the fundamental impossibility of universal concept verification. These results complement our earlier findings by showing that even with ideal implementation of our framework, certain conceptual properties of generative models may remain fundamentally unverifiable through probing.

### 3.1 IMPOSSIBILITY OF VERIFYING FINE-GRAINED CONCEPTS

We begin by demonstrating that for high-dimensional internal representations, it is impossible to verify the presence of all "fine-grained" concepts using a finite number of samples.

Fine-grained concepts, as defined in our work, represent properties that depend on precise patterns in the representation space. Technically, these concepts partition the d-dimensional space into $2^d$ orthants, with each concept corresponding to a specific sign pattern across all dimensions.

Intuitively, one can think of fine-grained concepts as very specific properties that require examining all dimensions of the representation space simultaneously. For example, in a language model, a fine-grained concept might be "formal technical writing with positive sentiment about mathematics" - a property that requires many subtle features to align in specific ways. In an image model, it might be "portrait photo with soft lighting from the left and a slight smile" - again requiring precise combinations of many features.

The key insight is that these concepts are "fine-grained" because they require exact matches across many dimensions, making them fundamentally hard to verify with any finite sampling strategy. This matches real-world intuitions about why it's hard to verify that generative models consistently capture very specific combinations of properties. We defer a technical definition to the proof of the following theorem in Appendix A.

**Theorem 2** (Fine-Grained Concept Verification). *Let $G : \mathcal{X} \to \mathcal{Y}$ be a generative model with a $d$-dimensional internal representation function $h : \mathcal{X} \to \mathbb{R}^d$. There exists a family of "fine-grained" concepts $\mathcal{C}$ such that for any probe $P$ using less than $O(2^d)$ samples, there exist two generative models $G_1$ and $G_2$ that are indistinguishable by $P$, but differ in whether they express a concept $c \in \mathcal{C}$.*

The key insight here is that in high-dimensional spaces, there are exponentially many "regions" (orthants in this case) where a concept could be present. With a limited number of samples, a probe is bound to miss some of these regions, allowing for the construction of models that differ in these unobserved areas.

This result suggests that for models with high-dimensional internal representations, it's practically impossible to verify all fine-grained concepts using a finite number of samples. In practice, probes might miss important but rare concepts in the representation space, and two seemingly identical models (from the probe's perspective) might have significantly different behaviors in certain regions - this can make auditing properties of such models hard (Bhattacharjee & von Luxburg, 2024).

## 3.2 Limitations of Linear Probes for Nonlinear Concepts

While our earlier results focused on linear probes for linearly separable concepts, we now explore their limitations when dealing with nonlinear concepts.

**Theorem 3** (Limitations of Linear Probes). *Let $G : \mathcal{X} \to \mathcal{Y}$ be a generative model with a $d$-dimensional internal representation function $h : \mathcal{X} \to \mathbb{R}^d$. Assume the components of $h(x)$ are independent and symmetrically distributed around zero. There exists a family of nonlinear concepts $\mathcal{C}_k$ of complexity $k$ defined over $\mathbb{R}^d$, such that any linear probe $P$ requires $O\left(\frac{k \log d + \log \frac{1}{\delta}}{\epsilon^2}\right)$ samples to reliably detect concepts from $\mathcal{C}_k$ within an error of $\epsilon$ with probability at least $1 - \delta$. Moreover, even with an infinite number of samples, a linear probe cannot perfectly capture these nonlinear concepts.*

This theorem reveals a duality in linear probes: efficiency in sampling versus limitations in expressiveness. While they can scale to high-dimensional spaces relatively efficiently, their inability to capture nonlinear relationships persists regardless of sample size. This has significant implications for probing strategies in deep learning, highlighting the need to balance the simplicity and interpretability of linear probes against the potential necessity for more expressive, nonlinear probing techniques when investigating complex concepts.

## 3.3 Impossibility of Universal Concept Verification

We now extend our analysis beyond specific probe types to consider the limitations of any fixed probing strategy.

**Theorem 4** (Universal Concept Verification). *Let $G : \mathcal{X} \to \mathcal{Y}$ be a generative model with a $d$-dimensional internal representation function $h : \mathcal{X} \to \mathcal{Z}$, where $\mathcal{Z} \subset \mathbb{R}^d$ is compact. For any fixed probing strategy $P$ with sample size $m$, there exists a concept $c : \mathcal{Z} \to \{0, 1\}$ that is strongly present in the model's internal representations but cannot be reliably detected by $P$.*

This result leverages the fact that any fixed probing strategy, with its limited sampling, can only distinguish among a finite number of concepts, while the space of possible concepts is much larger. It demonstrates the existence of strongly present yet undetectable concepts, highlighting fundamental limitations in our ability to comprehensively verify the internal representations of complex models using any fixed probing strategy.

This result is reminiscent of the "No Free Lunch" theorems in optimization and machine learning. It suggests that no single probing strategy can be universally effective, echoing limitations found in other areas of information theory and statistical learning.

## 3.4 Discussion

The apparent tension between our earlier sample complexity guarantees and later impossibility results reveals important insights about probing methodology. While Proposition 1 establishes polynomial sample complexity for concept detection, Theorems 2-4 demonstrate fundamental limitations. This seeming contradiction can be reconciled by examining their different contexts and assumptions.

First, these results address fundamentally different concept classes. Our early sample complexity bounds focus on linearly separable concepts, forming a restricted but learnable subset of all possible concepts. In contrast, our later theorems deal with increasingly complex concept spaces, from fine-grained patterns to nonlinear relationships, culminating in the full space of possible concepts. This progression demonstrates how sample complexity requirements escalate dramatically with concept complexity.

The results also highlight a crucial trade-off in probe expressiveness. Linear probes offer reliable learning guarantees but limited detection capability, while more expressive probes can capture complex concepts but face fundamental limitations in high-dimensional spaces. Together, these findings define the boundaries of what is and isn't possible with probing strategies.

These theoretical results suggest that while probing can provide valuable insights about certain concepts in generative models, it cannot offer complete guarantees about all possible concepts a model might encode. This inherent limitation points to the importance of considering probing as one tool

within a broader framework of interpretability methods. Future work may explore adaptive or iterative probing methods, the incorporation of domain-specific knowledge, and the complementary use of other analysis techniques such as adversarial testing or causal interventions.

## 4 RELATED WORKS

Interpreting deep learning models has been a significant research focus, with probes or diagnostic classifiers being a common approach. (Alain & Bengio, 2018) introduced using linear classifiers to probe hidden layers. In NLP, (Conneau et al., 2018) and (Tenney et al., 2019) applied probing to models like BERT, revealing layered learning of linguistic features. For computer vision, (Zeiler & Fergus, 2014) and (Bau et al., 2017) developed methods to visualize and quantify interpretability of convolutional networks.

However, limitations of probes have been debated. (Zhang et al., 2017b) showed networks could fit random labels, raising overfitting concerns. (Hewitt & Liang, 2019) argued high probe performance doesn't necessarily imply structural alignment with probed concepts. (Voita & Titov, 2020) and (Pimentel et al., 2020) cautioned against overinterpretation of probing results.

Our work extends this research by providing a formal framework ala (Dasgupta et al., 2022; Yadav et al., 2023) that defines probe use and rigorously analyzes their limitations, offering theoretical foundations for understanding when and why probes might fail to reveal meaningful insights.

## 5 CONCLUSION

We have presented a formal framework for explainability in generative models using probes, establishing a structured approach to analyze concept encoding within models. Our theoretical results highlight significant limitations in probing strategies, particularly in high-dimensional spaces and with nonlinear concepts. We showed that while linear probes are efficient and interpretable, they may fail to detect complex, nonlinear relationships, while more expressive probes can overfit to random labels.

Our findings suggest cautious application and interpretation of probing results. Additionally, there has been some recent work which argues for providing white box access to auditors for checking properties of ML models (Bhattacharjee & von Luxburg, 2024; Casper et al., 2024). Our work adds to that chorus while also suggesting that even white access (at least via probing) might not be enough in generative settings - there is a need to clarify what type of access is needed for efficient and rigorous audits.

Our work also suggests the use of complementary methods, such as causal interventions or alternative interpretability techniques, for a more comprehensive understanding of generative models. Future work may explore adaptive probing strategies or develop methodologies balancing probe expressiveness with reliability to address the challenges identified in our analysis.

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

# A PROOFS

This sections gives detailed proofs of all theorems in this paper.

## A.1 PROOF OF PROPOSITION 1

*Proof.* We consider training probes for all layers, validating them, and ensuring overall confidence and error bounds.

1. **Training Probes:** For each layer $l$, the VC dimension of the linear probe $p_{c,l}$ is $d_l + 1$. To achieve a generalization error of $\epsilon/3$ with confidence $1 - \delta/(3L)$ for each of the $L$ probes, we require:

$$n_{\text{train}} = O\left(\frac{1}{\epsilon^2}\left(d_{\max} + \log\frac{L}{\delta}\right)\right)$$

samples, where we use $d_{\max}$ to upper-bound all $d_l$. Note that we can use the same training samples for all layers, as we extract $h_l(x_i)$ for each $l$ from the same inputs $x_i$.

2. **Validating Probes:** To estimate the true error of each probe within $\epsilon/3$ with confidence $1 - \delta/(3L)$, we need:

$$n_{\text{val}} = O\left(\frac{1}{\epsilon^2}\log\frac{L}{\delta}\right)$$

validation samples, which can be shared across all probes.

3. **Ensuring Overall Confidence:** We allocate confidence levels: $\delta_{\text{train}} = \delta_{\text{val}} = \delta/(3L)$ for each probe's training and validation, and $\delta/3$ for the selection process. By the union bound, the probability of any failure is at most $\delta$.

The total sample complexity is the sum of training and validation samples:

$$n = n_{\text{train}} + n_{\text{val}} = O\left(\frac{1}{\epsilon^2}\left(d_{\max} + \log\frac{L}{\delta}\right)\right)$$

This sample complexity ensures that we can train probes across all $L$ layers, validate their performance, and select the best one while maintaining the desired error bound $\epsilon$ and confidence level $1 - \delta$ across all layers. $\square$

## A.2 PROOF OF THEOREM 1

*Proof.* We prove each part of the theorem separately:

**Part 1: Existence of a probe with perfect training accuracy**

Consider $n = \Omega(d)$ training samples $\{(h_l(x_i), c(x_i))\}_{i=1}^n$, where $h_l(x_i) \sim \mathcal{N}(0, I_d)$ and $c(x_i) \in \{0, 1\}$ are random labels independent of $h_l(x_i)$. Let the probe $p \in \mathcal{P}$ be a sufficiently expressive neural network, such as a multi-layer perceptron (MLP) with width $w = \Omega(n)$ and depth $k = \Omega(\log(n/\epsilon))$.

By the universal approximation theorem (Cybenko, 1989) and the results from Zhang et al. (Zhang et al., 2017a) on the ability of neural networks to fit random labels, we know that for any $\epsilon > 0$, there exists a neural network architecture that can achieve a training error less than $\epsilon$ on any labeling of the data, including random labels. Specifically, Zhang et al. showed that a neural network with $\Omega(n)$ parameters can express any function on a sample of size $n$.

Therefore, we can find a probe $p \in \mathcal{P}$ such that $p(h_l(x_i)) = c(x_i), \forall i = 1, \ldots, n$, achieving perfect training accuracy. The high dimensionality $d$ ensures that the network has sufficient capacity to memorize the training data.

**Part 2: Expected test accuracy**

Since $c$ is a random labeling independent of $h_l(x)$, the true relationship between $c$ and $h_l(x)$ is random. For a new sample $x' \sim D$, we need to show that $\mathbb{P}(p(h_l(x')) = c(x')) \leq 0.5 + \epsilon$.

Let $f$ be the function implemented by the neural network probe $p$. Despite fitting the training data perfectly, $f$ has no information about the true random labeling function for points outside the training set. For any new point $x'$, the probability that $f(h_l(x'))$ matches the random label $c(x')$ is 0.5.

By the law of large numbers, as the number of test samples increases, the test accuracy converges to 0.5. For any $\epsilon > 0$, we can bound the probability of the test accuracy exceeding $0.5 + \epsilon$ using Hoeffding's inequality:

$$\mathbb{P}(\text{PERF}_{\text{test}}(p, h_l, c) > 0.5 + \epsilon) \leq \exp(-2m\epsilon^2)$$

where $m$ is the number of test samples. By choosing $m = O(\frac{1}{\epsilon^2} \log \frac{1}{\delta})$, we ensure that the probability of the test accuracy exceeding $0.5 + \epsilon$ is at most $\delta$.

The high dimensionality $d$ contributes to this phenomenon by allowing the probe to perfectly fit the training data without capturing any true relationship, leading to poor generalization.

**Part 3: Mutual information**

Theoretically, since $c$ is independent of $h_l(x)$, the true mutual information $I(c; h_l) = 0$. To bound the empirical mutual information estimated from finite samples, we use the relationship between mutual information and KL divergence:

$$I(c; h_l) = \mathbb{E}_{c,h_l}[\log \frac{p(c,h_l)}{p(c)p(h_l)}] = \text{KL}(p(c, h_l)||p(c)p(h_l))$$

By Pinsker's inequality:

$$\text{KL}(p(c, h_l)||p(c)p(h_l)) \leq 2 \sup_A |p(c, h_l)(A) - p(c)p(h_l)(A)|^2$$

where the supremum is taken over all measurable sets $A$. The right-hand side can be bounded using concentration inequalities for high-dimensional Gaussians, specifically the Gaussian concentration inequality (Ledoux, 2001). For high-dimensional Gaussian random variables, the measure concentrates strongly around the mean, allowing us to bound the deviation between the empirical joint distribution and the product of marginals.

By choosing $D = O(\frac{1}{\epsilon^2} \log \frac{1}{\delta})$, we ensure that $I(c; h_l) \leq \epsilon$ with probability at least $1 - \delta$. The high dimensionality $d > D$ ensures that the concentration of measure phenomenon takes effect, leading to negligible mutual information between the concept and the representation. $\qquad\square$

### A.3 PROOF OF THEOREM 2

*Proof.* Let $\mathcal{Z} \subset \mathbb{R}^d$ be the space of internal representations produced by $h$. Without loss of generality, we can assume $\mathcal{Z}$ is bounded and rescale it to fit within $[-1, 1]^d$.

We define a family of fine-grained concepts $\mathcal{C}$ as follows: For each binary string $s \in \{0, 1\}^d$, define a concept $c_s : \mathcal{Z} \to \{0, 1\}$ as:

$$c_s(z) = \begin{cases} 1 & \text{if } \forall i, \text{sign}(z_i) = 2s_i - 1 \\ 0 & \text{otherwise} \end{cases}$$

Now, let $P$ be any probing strategy that uses $m < 2^d/(4d)$ samples. We will show that there exist two generative models that $P$ cannot distinguish, but which differ in whether they express some concept $c_s$.

Let $Z = \{z^{(1)}, \ldots, z^{(m)}\}$ be the set of internal representations sampled by $P$, where $z^{(i)} = h(x^{(i)})$ for some $x^{(i)} \in \mathcal{X}$.

For each $z^{(i)}$, let $s^{(i)} \in 0, 1^d$ be the binary string such that $\text{sign}(z_j^{(i)}) = 2s_j^{(i)} - 1$ for all $j$.

By the pigeonhole principle, since $m < 2^d/(4d)$, there must exist a binary string $s^* \in \{0, 1\}^d$ that does not appear in $\{s^{(1)}, \ldots, s^{(m)}\}$.

Now, we construct two generative models $G_1$ and $G_2$ as follows:

$G_1(x) = f(h(x))$ for some fixed function $f$.

$$G_2(x) = \begin{cases} f(h(x)) + \epsilon & \text{if } \forall i, \text{sign}(h(x)_i) = 2s_i^* - 1 \\ f(h(x)) & \text{otherwise} \end{cases}$$

where $\epsilon$ is a small perturbation vector.

Observe that $G_1$ and $G_2$ differ only in the orthant of $\mathcal{Z}$ corresponding to $s^*$, which is not sampled by $P$.

Therefore, $P$ cannot distinguish between $G_1$ and $G_2$.

However, $G_1$ does not express concept $c_{s^*}$, while $G_2$ does express $c_{s^*}$. This completes the proof. □

### A.4    PROOF OF THEOREM 3

*Proof.* Let $\mathcal{Z} = [-1, 1]^d$ represent the space of internal representations. We define the family of nonlinear concepts $\mathcal{C}_k$ as follows: For each subset $S \subseteq [d]$ of size $k$, define a concept $c_S : \mathcal{Z} \to \{0, 1\}$ as:

$$c_S(z) = \begin{cases} 1 & \text{if } \prod_{i \in S} z_i > 0 \\ 0 & \text{otherwise} \end{cases}$$

where $z = h(x)$ for some $x \in \mathcal{X}$.

Now, let $P$ be any linear probe. We will show that $P$ requires $O\left(\frac{k \log d + \log \frac{1}{\delta}}{\epsilon^2}\right)$ samples to reliably detect concepts from $\mathcal{C}_k$.

Let $Z = \{z^{(1)}, \ldots, z^{(m)}\}$ be the set of internal representations sampled by $P$. For any fixed concept $c_S \in \mathcal{C}_k$, the probability that a random point $z$ satisfies $c_S(z) = 1$ is $2^{-k+1}$, accounting for both all-positive and all-negative cases. For large $k$, this is of the same order as $2^{-k}$, which we'll use for simplicity in our calculations.

Using Hoeffding's inequality (which applies since $c_S(z) \in \{0, 1\}$ and the samples are independent), for $P$ to estimate the presence of $c_S$ within an error of $\epsilon$ with probability at least $1 - \delta'$, we need:

$$m \geq \frac{1}{2\epsilon^2} \log \frac{2}{\delta'}$$

There are $\binom{d}{k} = O(d^k)$ different concepts in $\mathcal{C}_k$. To reliably detect all concepts in $\mathcal{C}_k$, we apply a union bound over all these concepts. Setting $\delta' = \frac{\delta}{\binom{d}{k}}$, we get:

$$m \geq \frac{1}{2\epsilon^2} \log \frac{2\binom{d}{k}}{\delta} = O\left(\frac{k \log d + \log \frac{1}{\delta}}{\epsilon^2}\right)$$

Now, we show that even with infinite samples, a linear probe cannot perfectly capture all concepts in $\mathcal{C}_k$.

Let $w \in \mathbb{R}^d$ be the weight vector of the linear probe. For the probe to correctly classify a point $z$ for concept $c_S$, we must have:

$$\text{sign}(\langle w, z \rangle) = \text{sign}\left(\prod_{i \in S} z_i\right)$$

However, this equality cannot hold for all points in $\mathcal{Z}$ unless $w$ has non-zero components only in the dimensions indexed by $S$. Since there are $\binom{d}{k}$ different concepts, no single linear probe can correctly classify all points for all concepts in $\mathcal{C}_k$.

Therefore, even with infinite samples, a linear probe cannot perfectly capture all concepts in $\mathcal{C}_k$, completing the proof. □

### A.5    PROOF OF THEOREM 4

*Proof.* Let $P$ be any fixed probing strategy that samples $m$ points from $\mathcal{Z}$. We define a concept to be "strongly present" if it is true on at least half of $\mathcal{Z}$ by measure.

First, we note that the space of all possible concepts $c : \mathcal{Z} \to 0, 1$ is uncountably infinite, as it includes all measurable subsets of $\mathcal{Z}$. However, $P$, with its finite sample size $m$, can only distinguish among a finite number of concepts.

Let $Z = \{z^{(1)}, \dots, z^{(m)}\}$ be the set of points in $\mathcal{Z}$ sampled by $P$. These points partition $\mathcal{Z}$ into at most $2^m$ equivalence classes, based on the binary labeling of the sampled points. Consequently, $P$ can distinguish at most $2^m$ different concepts.

We now construct a concept $c$ that is strongly present but undetectable by $P$. Let $\mu$ be the Lebesgue measure on $\mathcal{Z}$. We can find a set $A \subset \mathcal{Z}$ such that: $\mu(A) \geq \frac{1}{2}\mu(\mathcal{Z})$ $A \cap Z = \emptyset$

Such a set $A$ exists because $Z$ is a finite set of points, which has measure zero in $\mathcal{Z}$. We then define $c$ as:

$$c(z) = \begin{cases} 1 & \text{if } z \in A \\ 0 & \text{otherwise} \end{cases}$$

By construction, $c$ is strongly present in $\mathcal{Z}$, as it is true on at least half of $\mathcal{Z}$ by measure.

Now, let $\hat{c}$ be any concept that $P$ outputs based on its samples. The error probability of $P$ in detecting $c$ is:

$$\Pr_{z \sim \text{Uniform}(\mathcal{Z})}[c(z) \neq \hat{c}(z)] \geq \Pr_{z \sim \text{Uniform}(\mathcal{Z})}[z \in A] \geq \frac{1}{2}$$

This error probability is significant, demonstrating that $P$ cannot reliably detect $c$. To embed $c$ in the generative model $G$, we can consider a modified model $G'$ where:

$$G'(x) = (G(x), f(h(x)))$$

where $f : \mathcal{Z} \to [0, 1]$ is a continuous function approximating $c$, such that $f(z) > \frac{1}{2}$ if and only if $c(z) = 1$. Such an $f$ exists by the Urysohn lemma, since $A$ is closed in the compact space $\mathcal{Z}$. This construction ensures that $c$ is strongly present in the internal representations of $G'$ and correlates with its output, yet remains undetectable by the probe $P$. $\qquad\square$

## B  ADDITIONAL EXAMPLES OF GENERATIVE MODELS

To further illustrate the versatility of our framework, we provide additional examples of generative models and demonstrate how they fit within our formalization.

### B.0.1  GENERATIVE ADVERSARIAL NETWORKS (GANS)

**Generative Model**   The generative model $M$ is the generator component of a GAN (Goodfellow et al., 2014), which produces synthetic data samples (e.g., images) from latent vectors $z \in \mathcal{Z}$.

**Hidden Representations**   The hidden representation at layer $l$ within the generator is $h_l(z) \in \mathcal{H}_l \subseteq \mathbb{R}^{d_l}$, representing the activations at that layer when processing a latent vector $z$.

**Concepts**   Concepts $c \in \mathcal{C}$ may include visual features such as edges, textures, shapes, or high-level attributes like facial expressions in generated images.

**Probes**   Probes $p_{c,l}$ are classifiers trained to detect the presence of concept $c$ in the hidden representation $h_l(z)$. For instance, a probe could determine whether a certain texture pattern is being represented at a particular layer of the generator.

### B.1  VARIATIONAL AUTOENCODERS (VAES)

**Generative Model:**  The generative model $M$ is a Variational Autoencoder (VAE) (Kingma & Welling, 2014), which consists of an encoder $E$ and a decoder $D$. The encoder maps input data $x$ to a latent representation $z$, and the decoder reconstructs the data from $z$.

**Hidden Representations:**  For an input $x \in \mathcal{X}$, the hidden representation at layer $l$ can be either within the encoder or decoder networks. We denote it as $h_l(x) \in \mathcal{H}_l \subseteq \mathbb{R}^{d_l}$, corresponding to the activations at layer $l$.

**Concepts:** Concepts $c \in \mathcal{C}$ may include features such as digit identity in handwritten digit datasets, facial attributes in face datasets (e.g., "smiling", "wearing glasses"), or more abstract properties like symmetry or style.

**Probes:** Probes $p_{c,l}$ are classifiers trained to detect the presence of concept $c$ in the hidden representation $h_l(x)$. For instance, a probe might predict whether a digit is even or odd based on the latent representation $z$ or other hidden layers.

## B.2 RECURRENT NEURAL NETWORK LANGUAGE MODELS

**Generative Model:** The generative model $M$ is a Recurrent Neural Network (RNN) language model, such as an LSTM (Hochreiter & Schmidhuber, 1997), which generates text by predicting the next word in a sequence based on the previous words.

**Hidden Representations:** For a sequence of words $x = (x_1, x_2, \ldots, x_t)$, the hidden representation at time step $t$ and layer $l$ is $h_l^{(t)} \in \mathcal{H}_l \subseteq \mathbb{R}^{d_l}$, representing the RNN's state at that point.

**Concepts:** Concepts $c \in \mathcal{C}$ could include grammatical number (singular or plural), tense (past, present, future), named entity types (person, location, organization), or discourse-level features like topic or sentiment.

**Probes:** Probes $p_{c,l}$ are trained to detect whether a particular concept $c$ is represented in $h_l^{(t)}$. For example, a probe might predict the tense of the current verb or whether the current word is part of a named entity.

## B.3 AUTOREGRESSIVE IMAGE GENERATORS

**Generative Model:** The generative model $M$ is an autoregressive image generator like Pixel-CNN (Oord et al., 2016), which models the distribution of pixel intensities conditioned on the values of previous pixels.

**Hidden Representations:** For an image $x \in \mathcal{X}$, the hidden representation at layer $l$ and pixel position $(i, j)$ is $h_l^{(i,j)} \in \mathcal{H}_l \subseteq \mathbb{R}^{d_l}$, representing the model's state before predicting the pixel at $(i, j)$.

**Concepts:** Concepts $c \in \mathcal{C}$ may include local image features like edges or textures, as well as higher-level properties such as object presence (e.g., "contains a tree") or scene type (e.g., "urban", "rural").

**Probes:** Probes $p_{c,l}$ are trained to detect the presence of concept $c$ in $h_l^{(i,j)}$. For instance, a probe could predict whether the current pixel is part of an object boundary or belongs to a particular semantic category.

## B.4 TRANSFORMER-BASED MUSIC GENERATION MODELS

**Generative Model:** The generative model $M$ is a Transformer-based model for music generation, such as Music Transformer (Huang et al., 2019), which generates musical sequences by attending to previous notes.

**Hidden Representations:** For a musical sequence $x = (x_1, x_2, \ldots, x_t)$, the hidden representation at layer $l$ and time step $t$ is $h_l^{(t)} \in \mathcal{H}_l \subseteq \mathbb{R}^{d_l}$, representing the model's processing of the sequence up to time $t$.

**Concepts:** Concepts $c \in \mathcal{C}$ might include musical properties such as chord progression, rhythm patterns, key signatures, or stylistic elements like genre or mood.

**Probes:** Probes $p_{c,l}$ are trained to detect whether a concept $c$ is present in $h_l^{(t)}$. For example, a probe could predict the current key signature or identify if a certain rhythmic pattern is being followed.

