# OpenReview forum: "The Probe Paradigm: A Theoretical Foundation for Explaining Generative Models"
_NeurIPS.cc/2024/Workshop/SafeGenAi — SafeGenAi Poster_

### Official Review · Reviewer_XsE3 · 2024-10-09
**Formal framework and theorems for informed application of  probing to generative model internal representations**

**Rating:** 6
**Confidence:** 3

**Review:**

Thank you for submitting to the NeurIPS SafeGenAI workshop! Your work identifies a critical but often overlooked gap in the application of high-performing, yet frequently uninterpreted, probing classifiers—a topic of growing importance in AI safety and security research.

First, I encourage you to take a more assertive stance on the need for researchers and peer reviewers to expect basic interpretability checks and common-sense validations in both linear and MLP-based probes. The field currently has a proliferation of high-performing linear probes with minimal interpretability assessments, resulting in a gap similar to the state of wildlife biology literature in the late 2000s, which lacked such foundational checks (see [2007 example](https://wildlife.onlinelibrary.wiley.com/doi/10.2193/2006-285)). While your paper emphasizes the interpretability of MLP probes, it is equally essential to apply interpretability measures to simpler linear probes to ensure their relevance and reliability.

Secondly, while your framework and its implications are valuable, the paper’s narrative would benefit from greater cohesion. Initially, it appears to address the need for interpretability in probes; however, it shifts toward presenting a formal framework for multi-level concept attribution, with occasional oscillation between the two themes. A more unified central message could enhance the paper’s clarity and impact.

**More specific and technical feedback:**
- *Definition of "fine-grained concepts":* This term lacks clarity. For example, does it encompass subjective notions like "honesty" or "happiness," as explored in the RepEng paper?
- *Have you considered inverse probing, where the probe aims to identify the absence rather than the presence of concepts?
- The statement "linear probes have a lower risk of overfitting but may miss complex concept encodings" could be expanded. For example, why do linear probes perform well in detecting concepts that shouldn’t be linearly separable? A recent paper shows that even a simple logistic regression can achieve high ROC AUC when detecting prompt injections ([example](https://arxiv.org/abs/2406.00799)).
- Theorem 1 assumes hidden representations drawn from a standard normal distribution, simplifying the mathematics but potentially limiting applicability to real-world generative model representations, which are often structured or clustered. Is this assumption realistic in your view?
- Strengthen your theoretical insights by empirically evaluating them. For instance, what evidence supports your claims about sample complexity and overfitting?
-  Proposition 2 states that a maximum of *d* independent concepts can be detected by linear probes in a *d*-dimensional space, based on hyperplane independence. This overlooks cases where concepts may overlap or correlate within the representations yet remain linearly separable. Additionally, superposition, where multiple concepts are encoded within overlapping subspaces, might challenge strict independence as the sole criterion for detection.
- Theorem 4 posits that “any fixed probing strategy with sample size *m*” cannot detect certain concepts with certainty, suggesting a no-free-lunch scenario. Can you clarify whether this limitation is due to concept complexity or dimensional constraints?
- *Minor issues:* Finally, some minor typographical errors detract from readability, such as inconsistent hyphenation (e.g., “explainability methods” vs. “explain-ability methods”) and a misspelling in the abstract (“presensce” instead of “presence”).

---

### Official Review · Reviewer_qfMH · 2024-10-10
**Nice paper studying theoretical properties of probing**

**Rating:** 7
**Confidence:** 3

**Review:**

The overall construction of the paper is nice. Even though the work is very theoretical I think it could benefit with some concrete real world or synthetic examples of the outcomes of the theoretical results. Furthermore, its not clear why model-agnostic framework is the best choice for studying generative models as they all contain very different hidden states i.e diffusion models are tasked with reconstructing noised representations at a certain hidden states, whereas layers within an LLM may focus on different aspects of a piece of text which to me seems very different and will therefore have varying implications with this papers theoretical results.

Strengths:
- Clearly defined problem statement and clearly proposed framework
- Solid theory which is straightforward to follow with good description about the results

Weaknesses:
- Examples for generative model types where the theory leads to results in real-world/synthetic set ups
- Clear motivation for the use of Model-agnostic framework being a preferred choice, I believe it is powerful for a framework to be model agnostic but it shouldn't oversimplify the definition of the different types of generative models to fit within the framework.
- More explanation on when the size of $d$ becomes an issue, i.e how high dimensional ? when can we expect this in some common models.

---

### Official Review · Reviewer_mLeg · 2024-10-12
**This manuscript provides a clear formal framework for explainability in generative models using probes, but it would benefit from a case study with popular models and a more thorough discussion of related works. Minor revisions are needed for formatting issues.**

**Rating:** 5
**Confidence:** 3

**Review:**

This work explores the theoretical explainability of generative models using probes, presenting a versatile framework with accompanying proofs. Some analysis highlights the limitations of the proposed approach.

Overall, the manuscript is well-structured, with a particularly clear definition of the formal framework. The section mapping LLMs and Diffusion models is highly useful. However, while the propositions are straightforward, it would strengthen the paper to include a case study with popular models to further demonstrate the applicability of the framework.

The discussion of related works is not extensive enough. Please consider elaborating on existing probe-related studies and emphasizing how the proposed framework differentiates itself from prior work.

Minor issues: Please update the header and correct 'Appendix ??' on Page 4, line 201.